# Development of Flowers Buds and Mixed Buds in the Dichasial Inflorescence of *Geranium koreanum* Kom. (Geraniaceae)

**DOI:** 10.3390/plants12183178

**Published:** 2023-09-05

**Authors:** Wanpei Lu, Zhongzhou Han, Qinghua Liu, Kuiling Wang, Qingchao Liu, Xuebin Song

**Affiliations:** College of Landscape Architecture and Forestry, Qingdao Agricultural University, Qingdao 266109, China

**Keywords:** *Geranium koreanum*, development of flower, mixed bud, inflorescence, morphology

## Abstract

Flower bud differentiation is of great significance for understanding plant evolution and ecological adaptability. The development of flower buds and mixed buds in the dichasial inflorescence of *Geranium koreanum* was described in this paper. The morphogenesis, surface structure, and organ morphology at different growth stages of *G. koreanum* buds were examined in detail using scanning electron microscope and stereo microscope. The development of mixed buds started from the flattened apical meristem. The stipule and leaf primordia arose first. Subsequently, the hemispherical meristem was divided into two hemispheres, forming a terminal bud and floral bud primordia, followed by lateral bud differentiation. The formation of the terminal and lateral buds of *G. koreanum* was sequential and their differentiation positions were also different. The floral bud primordia would develop into two flower units and four bracts. The primordia of a flower bud first formed the sepal primordia, then the stamen and petal primordia, and finally the pistil primordia. Compared to the stamen primordia, the growth of the petal primordia was slower. Finally, all organs, especially the petals and pistil, grew rapidly. When the pistil and petals exceeded the stamens and the petals changed color, the flower bud was ready to bloom.

## 1. Introduction

The genus *Geranium* L. (Geraniaceae Juss.) with about 325 species is distributed worldwide and new species are constantly being discovered [1,2,3,4]. A great number of geraniums are available to gardeners, and hundreds of cultivars have been used in gardens [5,6,7]. There are over 55 species in China [8], most of which have good ornamental value, such as *Geranium koreanum* Kom., while they are rarely used in gardens.

*G. koreanum,* also known as *G. tsingtauense*, is a leafy perennial plant native to northern China, the far east of Russia, and Korea [8]. The basal leaves are reniform and have three to five divisions. On the upright stem, there are abundant flowers with a diameter of about 3~4 cm, petals obovate, and colors ranging from bright to purple pink. The inflorescence is dichasial with 2-flowered cymules, similar to other species of the same genus [1]. Mature flowers contain five separate sepals and five petals. The position of the petals alternates with the position of the sepals. There are ten stamens divided into two whorls. The pistil has five carpels and five ovary locules. The style is connate at the base and divided into five stigmas at the top.

In the classification of *Geranium*, inflorescence types and flower characteristics are an important basis. The inflorescence types of *Geranium* are monochasial, dichasial, or scapigerous; some sepals are long and smooth, and some sepals are short and with longitudinal ribs; and some petals are large and hairy, and some petals are small and glabrous [9,10,11,12,13]. These different floral structures are related to the phylogenetic relationships between species [14,15,16]. Careful observation of the development of flowers can help understand their formation process. The macroscopic and microscopic morphological characteristics of flowers can play an important role in the classification of *Geranium* [17]. In addition, many studies have shown that the flower morphology of *Geranium* has a significant impact on flower evolution and pollination behavior [18,19,20,21]. However, detailed research on the early development of mixed buds and flower buds in *G. koreanum* has not been reported. In this paper, we observed the structure of mixed buds and flower buds of *G. koreanum* using stereo microscope and scanning electron microscope. By examining the shape and surface features of developing organs, the formation of mixed bud and flower bud was explored. These findings will provide basic theories for understanding the evolution of *G. koreanum* and promoting its cultivation, reproduction, and application.

## 2. Results

### 2.1. Dichasial Inflorescence of Geranium koreanum

The inflorescence of *G. koreanum* was dichasial. It was found that in early spring, from the first branch of underground sprouts, a structure of two mixed buds and one floral bud appeared in the middle of two leaves (Figure 1A). During the development of *G. koreanum* inflorescence, each node had two leaves, two branches, and a small inflorescence of two flowers (Figure 1B). *G. koreanum* started to germinate in March and reached its peak flowering period in August. During this growth process, their branching order can reach seven or more or less according to the individual growth conditions (Figure 1B).

In addition, we found that the flower buds on the first, second, and third order branches were usually unable to open, and they gradually withered in the later stages of growth. Regarding the flower buds on the fourth-order branches, depending on the growth status of the plant, some could open, while others could not. Based on the observed results, we obtained a schematic diagram of the inflorescence pattern of *G. koreanum* (Figure 1C).

Another interesting phenomenon was that although it was a typical dichasial cyme, the growth speed of its two branches on the same node was not the same. It was obvious that one branch exhibited the apical dominance of the terminal bud for growth while the other branch grew slower. In the schematic diagram, we used longer branches to represent faster growing branches, and shorter branches to represent slower growing branches. We found that if we only looked at the growth direction of the faster growing branches, their branching pattern was similar to a racemose branch, with slower developing branches arranged alternately on both sides (Figure 1B,C).

The number of graded branches in *G. koreanum* could reach seven or more. If calculated based on the open flower buds from the fifth order with two branches at the base, the number of *G. koreanum* flowers was astonishing. Due to different growth environments, each plant may have some differences, but it was evident that the number of flowers of *G. koreanum* was particularly large, with at least a dozen flowers per plant (Figure 1D).

### 2.2. Mixed Bud Morphology

The top part of the inflorescence of *G. koreanum* contained two branches, one floral bud, and two leaves (Figure 2A). The top part of branch was taken down for observation, and it could be clearly seen that two stipules wrapped around the leaves and buds at the base of the node (Figure 2B,C). After dissecting the bud by removing its stipules and leaves, it could be observed that it contained two buds and one floral bud (Figure 2D,E). The two buds, floral bud, and two leaves were arranged in a linear pattern inside the top bud (Figure 2C,D). One of the two buds inside was big and the other was small (Figure 2E). According to the observation of the inflorescence above, the growth speed of two branches on the same node was different: one was fast and the other was slow, and one had a clear the apical dominance. Among the two buds dissected at the top part of the branch, the larger one clearly grew faster than the smaller one. Therefore, we referred to the larger bud as the terminal bud, whereas the smaller bud was called the lateral bud. Continuing to dissect the buds by removing the stipules and leaves, it was found that they had the same structure, comprising a new terminal bud, a new lateral bud, and a new floral bud (Figure 2F). Obviously, the top buds of *G. koreanum* was a mixed bud.

### 2.3. Mixed Bud Organogenesis

In the section above, we observed under macroscopic anatomical observation that the top bud of *G. koreanum* was a mixed bud. We then observed its organogenesis process in the ultrastructural observation. The apical meristem initially appeared in a flat form (Figure 3A). The center of the apical meristem gradually rose, and the marginal primordium were gradually separated by a groove (Figure 3B,C). The edge primordia differentiated into leaf primordia and stipular primordia. The meristem grew into a square shape, and the leaf primordia and stipular primordia were becoming increasingly prominent with the leaf primordia formed at both ends of the long axis and the stipule primordia formed at both ends of the short axis (Figure 3C,D). The central part of the apical meristem grew upward, initially forming a dome shape, and then gradually divided into two parts (Figure 3D–F). The small portion developed into a new terminal bud (TB), while the large part formed a floral bud (Figure 3G,H). The primordia of the stipule grew rapidly, enveloping both the central growth point and the leaves (Figure 3I).

Then, the leaf primordia also initiated the differentiation of lobes, and the floral bud began to differentiate and form bracts (Figure 3G–I). At the same time, the terminal bud (TB) also began a new round of differentiation, just like the differentiation of the apical primordia in Figure 3B, in which the stipules and the leaf primordia differentiated at the edge of the primordia (Figure 3H,I). When the floral bud and the terminal bud primordia grew to a certain period, the lateral bud began to differentiate on the other side of the floral bud. When the lateral bud began to differentiate, it was flat (Figure 3J) and then became raised (Figure 3K), and it gradually differentiated into a complete lateral bud, which is akin to how the differentiation started in Figure 3A as well. Finally, a complete mixed bud was formed, consisting of two stipules, two young leaves, one lateral bud, one floral bud, and one terminal bud (Figure 2 and Figure 3L).

### 2.4. Floral Morphology

Flowers of *G. koreanum* grew in pairs. They were structured with pedicel, four bracts, two peduncles, and two complete flowers that were actinomorphic and that had five sepals, five petals, and ten stamens (Figure 4C,D). The sepals had hairs, longitudinal ribs, and mucro (Figure 4A–C). The petals were obovate, purple pink, and with opaque purple veins at the base. The petals and sepals of *G. koreanum* alternately grew (Figure 4D).

### 2.5. Floral Organogenesis

As the mixed buds developed, the floral buds of *G. koreanum* also developed together. During the process of mixed bud differentiation, both floral bud and terminal bud primordia differentiated simultaneously (Figure 3F–I). The floral bud primordium further differentiated into two flower units, including peduncles, bracts, and two complete flowers (Figure 4C). The first organ to appear in the floral bud primordium was bracts and the second was calyx, followed by androecium and corolla, and pistil was the last to form. We concluded the floral organogenesis of *G. koreanum* in order of their appearance.

#### 2.5.1. Bracts

After the apical meristem gradually developed to form stipule primordia and leaf primordia, the central meristem differentiated a floral bud primordium and a terminal bud primordium (Figure 3F). The floral bud primordium first developed two small bracts in the long axis direction, bract 1 and bract 2 (Figure 3G,H and Figure 5A). The central meristem of the bud differentiated into two hemispherical primordia, which would develop into two flowers (Figure 5A). The development of these two flower bud primordia was not synchronous, with the larger flower bud primordium developing first and gradually exhibiting sepal differentiation, while the other small flower bud primordium developed to form two other bracts (bracts 3 and 4) (Figure 5B). Bract 1 and bract 2 grew rapidly, and trichome appeared on their backs (Figure 5C). Bract 1 quickly covered the two flower buds, and only by removing it could the growth of the flower bud be observed (Figure 5D). At that time, bracts 3 and 4 were still very small, and differentiation of five sepals had been observed on the larger bud primordium, while sepal differentiation had not yet begun on the small flower bud primordium (Figure 5D). Bracts 1 and 2 then grew rapidly and covered the two flowers, and bracts 3 and 4 grew upwards, too, with trichomes on the back (Figure 5E). Finally, the four bracts were wrapped around the flower buds. Two flower buds, one large and one small, were located in the middle of the bracts (Figure 5E,F).

#### 2.5.2. Calyx

When bract 3 and bract 4 began to differentiate on the smaller flower bud primordium, the larger flower bud primordium had already begun to form sepals (Figure 5B). When the two flower bud primordia began to differentiate, they appeared in a hemispherical shape (Figure 5A), but when the sepal primordia began to differentiate, they appeared in an irregular polygonal shape (Figure 6A). When the first and second sepals differentiated from the direction parallel to the boundary between the two flower buds (Figure 6B), they were opposite to each other, and then the third and fourth sepals differentiated in their vertical direction (Figure 6C). Finally, the fifth sepal differentiated between the second and third sepals on the opposite side of the first sepal, and the central meristem presented a standard pentagon (Figure 6C,D).

The first sepal grew rapidly with morphological differentiation and the development of glandular hairs (Figure 6D). The other sepals began to develop and grew in the order of differentiation. When the largest sepal was removed, it was seen that the second sepal developed the fastest among the other sepals. The sepals were arranged in the order of differentiation from the outside to the inside (Figure 6E). The first sepal was on the outermost side, covering the third and fourth sepals; the second sepal was on the opposite side of the first sepal, covering its adjacent fourth and fifth sepals; and the fifth sepal was the last to develop and was arranged in the innermost layer. Finally, they developed into sepals with a pointed head, hairy glands, and rids tightly enveloping the internal structure (Figure 6F). As the sepals developed, their hairs increased more and more with a wide base, forming a structure that tightly wrapped the meristem in the middle (Figure 6G,H).

#### 2.5.3. Androecium and Corolla

When the differentiation of the five sepal primordia was completed, the central meristem in the flower bud became a regular pentagon, and it was ready to start the differentiation of stamens and petals (Figure 6C,D and Figure 7A). In our observation, we found that the organogenesis of stamens and petals was simultaneous.

First, the 10 stamen primordia began to differentiate simultaneously at the edge of the five-star shaped meristem primordia. When the stamen primordia began to differentiate on the meristem, they were faintly visible, forming 10 small protrusions (Figure 7B), followed by the formation of clearly visible circular protrusions (Figure 7C). The development of the 10 stamen primordia gradually produced different states. The development of the five stamen primordia located above the five corners was slower than that of the other five stamen primordia, which meant that these stamen primordia located between the five corners grew significantly faster (Figure 7D,E). Faster developing stamens would form the longer stamens, while slower developing stamens located above the five corners would develop into the shorter stamens (Figure 7F). When the long and the short stamens were clearly layered, the primordia began to differentiate into filaments and anthers (Figure 7F,G), with the filaments gradually becoming slender and the anthers gradually developing laterally. Finally, a complete filament structure and anther structure were formed, with the high and low stamens arranged alternately (Figure 7H,I).

The development of petal primordia was almost synchronous with the development of stamen primordia. Initially, when the sepal primordia formed 10 small protrusions, it differentiated under the facade of the five corners of the pentagonal primordium to form bulges (Figure 7B). The bulges below the five corners of the primordium gradually formed an obvious boundary with the stamen primordia above when the 10 stamen primordia gradually formed different states (Figure 7C–E). Then, the petal primordia developed from a full and round shape into a regular shape when the 10 stamen primordia began to layer (Figure 7F,G). Finally, the petal primordia formed a thin, nearly circular shape when the filament and anther structure developed on the stamen primordia (Figure 7H,I).

From the above observation, the results indicated that in the early stage of flower bud development, although the differentiation of stamens and petals was almost simultaneous, the development of stamens was significantly faster than that of petals. When the stamen morphology was fully developed, the size of the petals was still very small. In the later stage of development, the growth rate of the petals exceeded that of the stamens. The petals were initially transparent (Figure 8A) and gradually turned white (Figure 8B). Then, the veins at the base of the petals first turned purple (Figure 8C), and, finally, the entire petals turned purple and wrapped the stamens and pistil (Figure 8D,E).

#### 2.5.4. Pistil

During flower development, the formation of pistil was delayed compared to calyx, petals, and stamens. The central meristem of the flower bud was flat at the initial stage of stamen differentiation. With the formation of the stamen primordia, the central meristem formed a pentagonal shape again (Figure 7E). At that time, it was ready to start pistil differentiation. The pistil primordium appeared as a ridge along the margin of the pentagon (Figure 9A), and then grew upward at its five corners, like a mountain with five peaks (Figure 9B,C). Five peaks grew centripetally and eventually met (Figure 9D–I). As the outer part of the pistil primordium grew upwards like a mountain peak, its interior gradually differentiated into partitions and extended inward to the central point (Figure 9D,F), forming five carpels (Figure 9H,I). The five carpels appeared simultaneously and grew together. When five peaks met, the style was formed at the top and the ovary with five ventricles was formed at the bottom.

In the flower bud development of *G. koreanum*, the differentiation of the pistil was the last step. The formation of pistil morphology was much later than that of stamens. However, in the later stage of flower bud development, the pistil growth speed was almost as fast as that of the petals, faster than that of the stamens. After the top parts of the pistil primordium grew together, the length and width of the ovary increased, and the pistil grew upwards (Figure 10A,B). The ovary and style of the pistil were united, but the stigmas were not fused, and the detached stigmas gradually developed (Figure 10B–D). The trichomes in the ovary and style began to grow and became denser (Figure 10C–E). The length of the pistil gradually exceeded that of short and long stamens, and finally exceeded that of the stamen group before the flower opening (Figure 10C–E).

## 3. Discussion

### 3.1. Inflorescence and Origin of Flowers

Some species of *Geranium* have dichasium inflorescences, such as *G. jaramilloi* and *G. ocellatum* [3,22]. During the development of dichasial cymes, flower buds form at the top and two branches are formed on both sides [23]. In the process of the development of *G. koreanum* inflorescence, we observed that the apical meristem of *G. koreanum* differentiated simultaneously into a floral bud and a mixed bud at first, while the differentiation of the other mixed bud was later (Figure 3). In the later stage of growth and development, the growth rate of the first mixed bud was much faster than that of the flower bud and the second mixed bud. Moreover, according to the observation of development of the inflorescence, the growth speed of two branches on the same node was different––one fast and the other slow. The bud that developed first had a significant apical advantage (Figure 1 and Figure 3). Therefore, we referred to the first mixed bud as the terminal bud in the article. These results could provide some reference for us to further study the evolution and formation of *G. koreanum* dichasial inflorescences.

From the analysis of the previous results, we obtained the developmental process of the dichasial inflorescence of *G. koreanum* (Figure 11). The apical meristem initially appeared in a flat form (Figure 11A). Then, it formed the stipular primordia and the leaf primordia (Figure 11B), and followed the differentiation of the floral bud primordium and the terminal bud primordium (Figure 11C). Subsequently, the floral bud primordium began to form two bract primordia and two flower buds, and the terminal bud primordium began to form the new stipule and leaf primordia (Figure 11D). Finally, it was the differentiation of the lateral bud primordium and the other two bract primordia on the floral bud (Figure 11E). In this way, the mixed bud of the dichasial inflorescence of *G. koreanum* was formed. The developmental order of the mixed bud of *G. koreanum* was stipules and leaves, floral bud and terminal bud, and then lateral bud.

From the above process, it was found that the differentiation of *G. koreanum* inflorescences started from the differentiation of stipular and leaf primordia. Stipules are considered as outgrowths or appendages at the base of petioles, growing in an orderly manner on the stem nodes of flowering plants [24]. The research on the stipules had a long tradition, and there were rich achievements in the research on the origin of stipules. For example, some scholars believed that Dicotyledon stipules and Monocotyledon leaf tongues had different development origins [25]. A systematic study of stipules and colleters of mangrove Rhizophoraceae showed that the morphology correlated with the shape of their colleter aggregations [26]; the presence of stipules in the outer organs of apetala2-1 flowers was evidence of their leaf-like structure [27,28]; and other studies were related to the formation and evolution of stipules [24,29]. In our observation, the apical meristem of *G. koreanum* first formed a circle of divisions around its periphery, while differentiating into stipule and leaf primordia. Stipule primordia grew faster than the leaf primordia and quickly enveloped the entire mixed bud. The stipules quickly wrapped the apical meristem tissue before other organs had formed, playing an important protective role in the development of mixed buds like the other plant’s stipules [29,30,31]. Furthermore, the formation of stipules also played a role in functional zoning. With the formation of stipules, it meant the formation of an independent new branch, which would initiate the mechanism of development of new mixed buds. The formation of stipule may have an indicative effect on the formation of branch. These observations could provide some reference for studying the origin and function of *G. koreanum* stipules.

We found that in the early spring, the first branch development of *G. koreanum* followed this branching pattern. The top bud of the first branch of underground sprouts contained one floral bud in the middle and two branches on both sides. However, the early flower buds did not bloom normally. We found that the flower buds on the first, second, and third order branches were usually unable to open. The plants need to grow for a certain period of time before they bloom normally. The question of why the first flower buds cannot bloom requires us to continue observing and researching. It would be interesting to know why this happens and to know the commencement of flower development.

Additionally, it was interesting that the growth speed of two branches on the same node in *G. koreanum* was not the same. In the schematic diagram of the inflorescence pattern of *G. koreanum* (Figure 1C), we found that the branching pattern of *G. koreanum* was similar to a racemose branch [23]. The faster growing branches formed the apical dominance and the slower developing branches arranged alternately on both sides (Figure 1B,C). Moreover, through scanning electron observation, it was found that during the origin process of mixed buds, the terminal bud and the floral bud differentiated simultaneously, while the lateral buds differentiated on one side of the floral bud only after the terminal bud and the floral bud formed. This order and position are also in line with the definition of lateral buds. Although the terminal bud and the lateral bud of *G. koreanum* were located on the same node, their formation was sequential and their differentiation positions were also different. Why do the terminal and lateral buds of the dichasial inflorescence of *G. koreanum* not originate simultaneously? Continuing to study this issue should be very interesting.

### 3.2. Flower Structure and Organogenesis

*G. koreanum* contains relatively simple floral structures. *Geranium* species have five sepals, five petals, ten stamens in two whorls, and five carpels [8]. The flower structure of *G. koreanum* is also similar to that of other species of *Geranium* [18,32].

Based on our observations, we also obtained the development process of *G. koreanum* flower (Figure 12). The flower development of *G. koreanum* first formed five sepal primordia in sequence, then simultaneously differentiated into five petal primordia and ten stamen primordia, and finally formed the pistil primordia (Figure 11). The five sepals differentiated first, and they had an obvious sequence, which was similar to the sequence of other calyx development, such as Rhamnaceae and *Dipteryx alata* [33,34]. The order of bract and sepal initiation varies among the species, and these are related to the evolutionary relationships between species [34,35,36]. The succession of the five sepals may provide the morphological traits to study the sepal origin of *G. koreanum*.

During this process, the differentiation of the petals and stamens was different from the sequential differentiation of the sepals. The five petal primordia and ten stamen primordia differentiated simultaneously. The origin of the petals was believed to originate from bracts or stamens [37]. The petal primordia of *G. koreanum* differentiated simultaneously with the stamen primordia and should be of the same origin as the stamens. Their differentiation position was also determined. When sepal differentiation was completed, the top meristem was enclosed by the sepal primordium to form a pentagon shape, and then the differentiation position of the sepals, petals, and stamens was determined. At the time of initial differentiation, five sepals were enclosed on five sides of the pentagon, which corresponded to the differentiation of stamens in front of the sepals (antesepalous stamen), while petals are differentiated on five corners of the pentagon, which corresponded to the differentiation of stamen primordia in front of the petals (antepetalous stamens). These observations were similar to those of other plants of the same genus, such as *G. robertianum* [18].

The developments of petals were delayed after their initiation and lagged behind that of stamens in most Eudicots [23,38,39]. The early development of the petals of *G. koreanum* was also lagging behind that of the stamens. This could be related to the tangential division of petals and stamens that occurred simultaneously from these common primordia [38,40]. The elaborate structure of petals is often related to attracting insects for pollination [40,41,42,43,44], and petal color plays an important role in it [45,46]. The petals of *G. koreanum* were obovate, with hairs at the base, and each petal had five dark purple veins. We observed the appearance of veins in the early stages of petal development, and the discoloration of petals also started from the veins. These results may provide some information for understanding the fine development of petals in *G. koreanum*, but this study did not systematically observe the development of petal structure, and we need more specific and detailed work to explore it.

## 4. Materials and Methods

Early mixed buds and flower buds were collected for research from plants of *G. koreanum* cultivated in the experimental field of Qingdao Agricultural University. Fresh tissue was dissected under a stereo microscope (JM, Olympus, Tokyo, Japan) and photographed using a digital camera (MD30, Mshot, Guangzhou, China).

Other fresh tissues were preserved in FAA (formalin: acetic acid: 70% ethanol; 5:5:90 *v*/*v*) and then transferred to 70% ethanol, dissected under a stereo microscope, and then prepared for the next step. The specimens were dehydrated using ethanol series (70%, 85, 95% and 100%), and then using 50/50 ethanol/tert-butyl alcohol (TBA) and 100% TBA. The specimens were freeze dried using a JEOL JED-320 freeze drying device and then sputter-coated with gold using a JEOL JFC-1600 auto fine coater for examination under field emission scanning electron microscope (JSM-7500F, JEOL, Tokyo, Japan).

## 5. Conclusions

Our observation provides a series of morphological developmental progress to study the flower and inflorescence organogenesis of *G. koreanum*. The differentiation of mixed buds began with the differentiation of stipule and leaf primordia, followed by the differentiation of terminal bud and floral bud primordia, and, finally, the differentiation of lateral buds. The terminal bud and the lateral bud in the dichasial inflorescence of *G. koreanum* were located on the same node, but their formation was sequential and their differentiation positions were also different. The order of flower development of *G. koreanum* was sepals, stamens and petals, and pistil. The differentiation of the five sepals was sequential. The petals and stamens arose simultaneously and the early development of petals was delayed after initiation. The petals and pistil grew rapidly in later stages, surpassing stamens before flowering.

## Figures and Tables

**Figure 1 plants-12-03178-f001:**
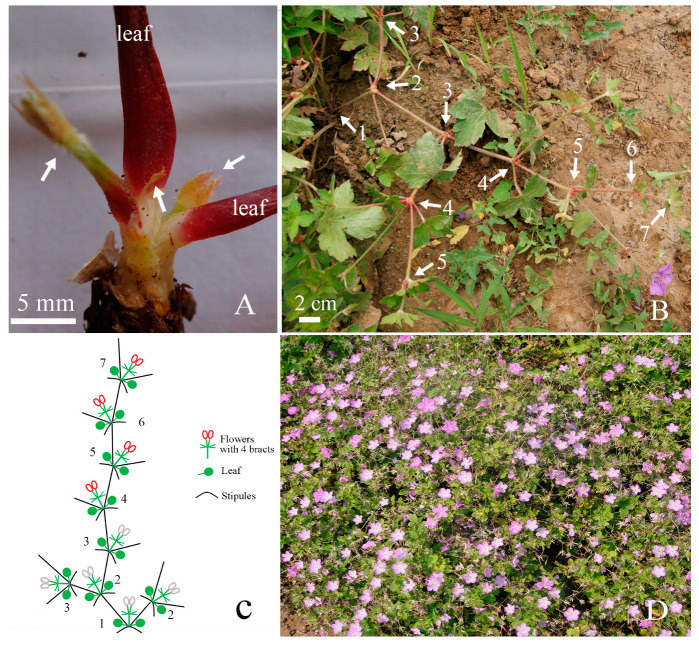
Inflorescence of *G. koreanum*. (**A**) The first branch of underground sprout. Arrows indicate two branches and one floral bud. (**B**) The inflorescence of *G. koreanum*. Numbers represent branching order. (**C**) The schematic diagram of the inflorescence pattern of *G. koreanum*. The red ring indicates that the flower is usually able to open, while the gray ring indicates that the flower is usually unable to open. Numbers represent branching order. (**D**) *G. koreanum* in its peak flowering period.

**Figure 2 plants-12-03178-f002:**
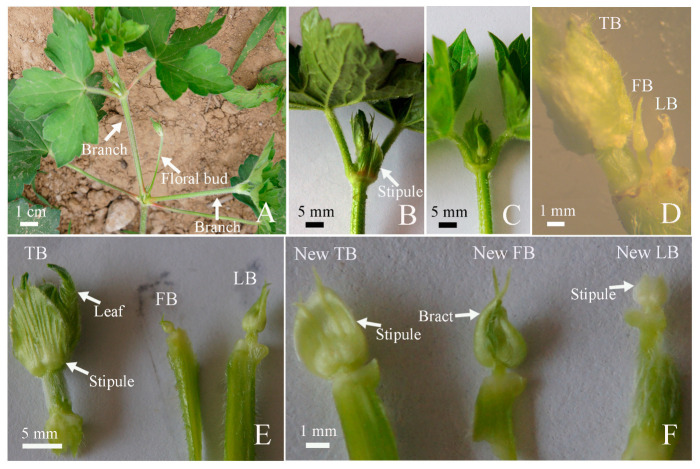
Mixed buds of *G. koreanum*. (**A**) The top part of inflorescence. Arrows indicate branches and flowers. (**B**) The top part of the branch. The arrow indicates a stipule. (**C**) The top part of the branch, with one stipule removed. (**D**) The top part of the branch, with leaves and stipules removed. (**E**) The terminal bud (TB), the floral bud (FB), and the lateral bud (LB) separated. (**F**) Anatomy of the terminal bud (TB) in Figure 3E. The new terminal bud, the new floral bud, and the new lateral bud separated from the TB, with the leaves and stipules removed.

**Figure 3 plants-12-03178-f003:**
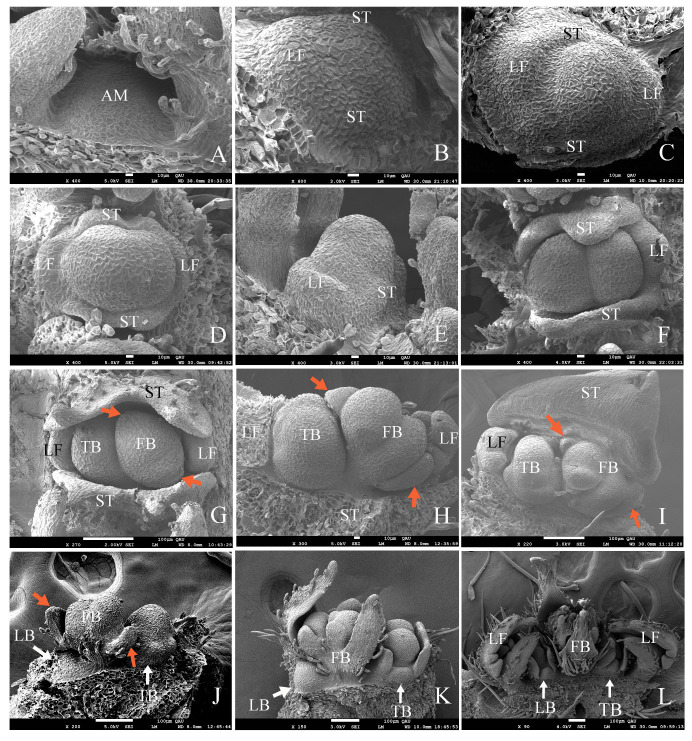
Scanning electron micrographs of mixed bud development in *G. koreanum*. (**A**) The apical meristem in a flat form. (**B**) The apical meristem that protruded upwards. Leaf primordia and stipular primordia began to differentiate at the edge of the primordia. (**C**) The center part of the apical meristem formed a hemisphere and the leaf primordia and stipular primordia were becoming increasingly prominent. (**D**,**E**) The apical meristem grew into a square shape, and the center part was significantly raised and began to differentiate. (**F**) The leaf primordia and stipule primordia grew upward. The center part differentiated into two distinct hemispheres, one of which would develop into a floral bud and the other into a terminal bud. (**G**) The stipule primordia grew rapidly with a glandular hair appearance. The floral bud primordium and the terminal bud primordium were separated. Two ends of the flower bud became longer and sharper, and two bract primordia began to differentiate. (**H**) The apical meristem with two stipules and one leaf removed. The two bract primordia had already formed. The terminal bud primordium grew into a square shape, and the center part was significantly raised and began to differentiate. The leaf primordia started three lobe differentiation. (**I**) The stipule enveloped the other primordia and one stipule was removed. (**J**) The lateral bud appeared in the form of a plane, with stipules and leaves removed. (**K**) The lateral bud enlarged as a hemisphere, and stipules and leaves were removed. (**L**) A completed mixed bud, including two young leaves, one lateral bud, one floral bud, and one terminal bud, with two stipules removed. LF, leaf; ST, stipule; FB, floral bud; AM, apical meristem; TB, terminal bud; LB, lateral bud. Red arrows indicate bracts. White arrows indicate LB and TB.

**Figure 4 plants-12-03178-f004:**
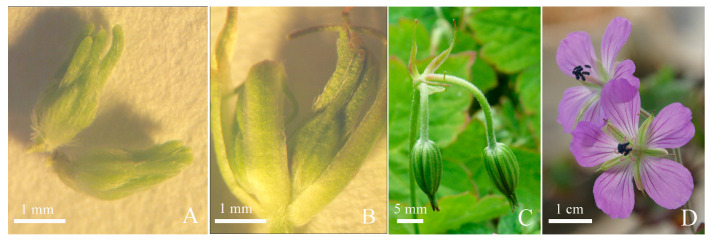
Flowers and floral buds of *G. koreanum*. (**A**) Small floral buds with bracts removed. (**B**) Small floral buds with bracts. (**C**) One pair of flower buds nearly open. (**D**) Flowers in blossom.

**Figure 5 plants-12-03178-f005:**
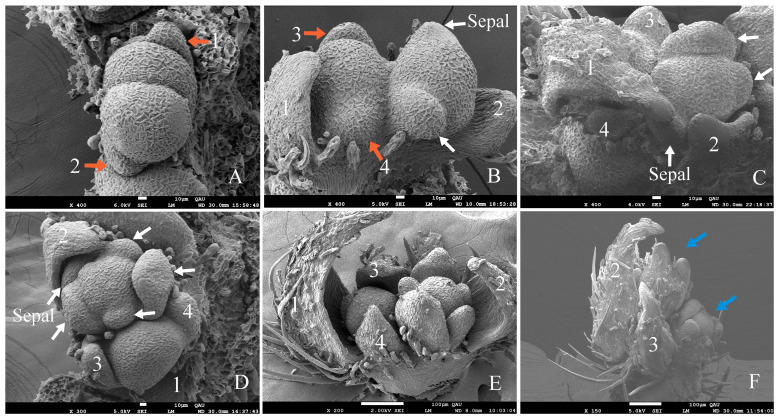
Scanning electron micrographs of bracts development in *G. koreanum*. (**A**) The floral bud primordium with outer bracts appearance, bract 1 and bract 2. (**B**) The lateral bract primordia emerged on the sides of the small flower primordium when the sepal primordia emerged in the bigger flower primordium. Bract 1 and bract 2 grew upwards with trichomes on the back. (**C**) Bracts 1 and 2 grew rapidly, but bracts 3 and 4 grew relatively slowly. Bract 1 lengthened and curved inward. (**D**) Bract 2 lengthened and curved inward with bract 1 removed. Differentiation of five sepals had been observed in the larger bud primordium. (**E**) Bracts 1 and 2 covered the two flower and bracts 3 and 4 grew upwards with trichomes on the back. (**F**) The bracts were wrapped around the flower buds, with bract 1 removed. Two flower buds, one large and one small, were located in the middle of the bracts. The numbers 1–4 represent the chronological order of bract development. Red arrows indicate bracts. White arrows indicate sepals. Blue arrows indicate flower buds.

**Figure 6 plants-12-03178-f006:**
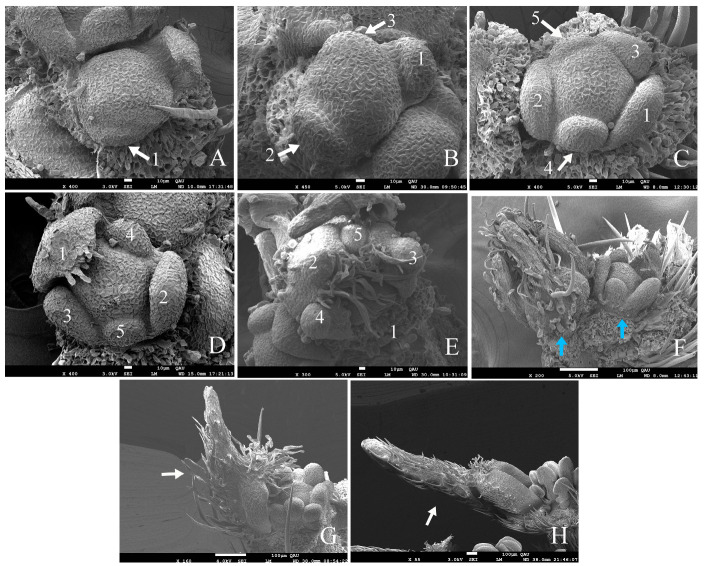
Scanning electron micrographs of calyx development in *G. koreanum.* (**A**) The flower bud primordium as an irregular polygon. (**B**,**C**) Sepal primordia appearing sequentially on the flower bud primordium. (**D**) The first sepal that began to cover the bud with dense trichomes emerged on the front tip. (**E**) The five sepals covered the entire apical meristem, with the first sepal removed. (**F**) Two flower buds. The five sepals of the larger flower bud had already differentiated and formed the rids and hairs, while the five sepals of the smaller flower bud had just formed. (**G**,**H**) Sepals at different stages of development. The numbers 1–5 represent the chronological order of sepal development. White arrows indicate sepals. Blue arrows indicate flower buds.

**Figure 7 plants-12-03178-f007:**
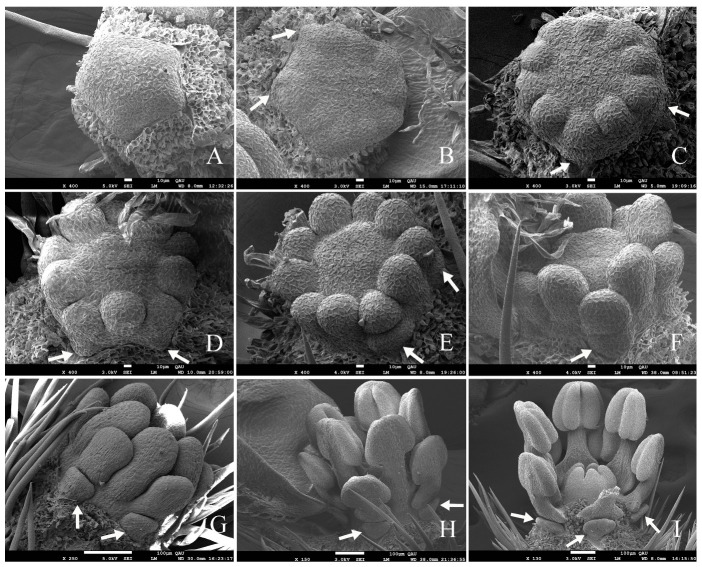
Scanning electron micrographs of androecium and petal development in *G. koreanum*. (**A**) Five-angled floral primordium, with sepals removed. (**B**) The ten stamen primordia faintly visible at the edge of the meristem and the five petal primordia barely visible at five angles. (**C**) The stamen and sepal primordia were clearly visible. (**D**,**E**) Gradually developing stamen primordia and petal primordia. (**F**) The long stamens grew faster than the short stamens. (**G**) The stamen primordia that became stalked and the petal primordia that separated from the short stamen primordia. (**H**) The stamen primordia that began to differentiate into filaments and anthers. (**I**) The stamens with filaments and anthers, with one short stamen and two long stamens removed. White arrows indicate petal primordia.

**Figure 8 plants-12-03178-f008:**
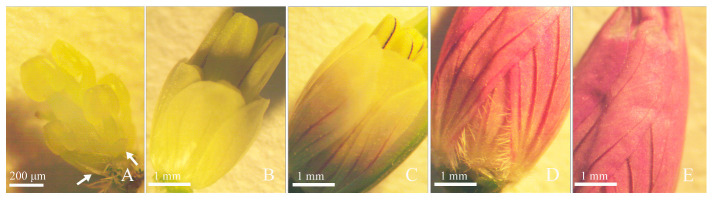
Corolla development. (**A**) Flower bud showing that the petals were still small compared to the stamens at the early stage of development. (**B**) Flower bud in which the petals had just reached the height of the short stamens. (**C**) Flower bud in which the petals surpassed the short stamens with slightly pale pink. (**D**,**E**) Flower bud ready to open in which the petals enveloped the stamens totally. White arrows indicate petals.

**Figure 9 plants-12-03178-f009:**
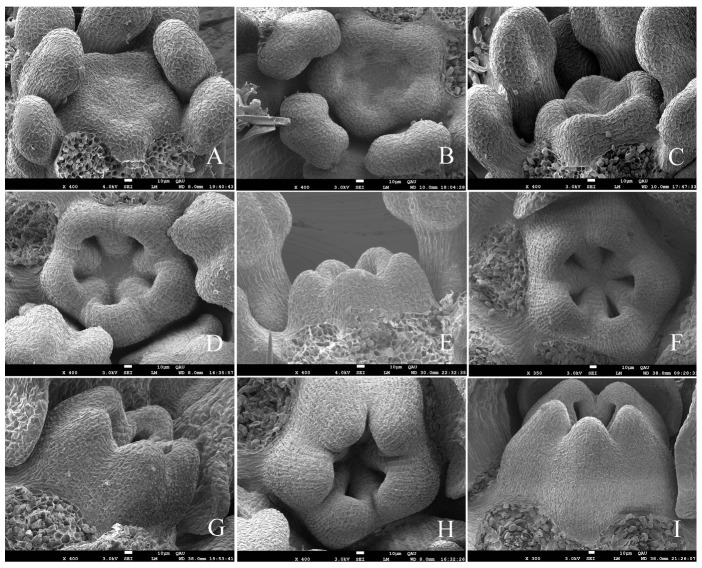
Scanning electron micrographs of pistil development in *G. koreanum*. (**A**) The pistil primordium appeared as a ridge along the margin of the pentagon and arose slightly higher in the angles. (**B**,**C**) The pistil primordium grew vertically. (**D**,**E**) The pistil primordium grew vertically and inward to form a compartment, forming the carpels and ventricles. (**F**,**G**) The five carpels gradually grew together. (**H**,**I**) The pistil whose carpels were about to meet.

**Figure 10 plants-12-03178-f010:**
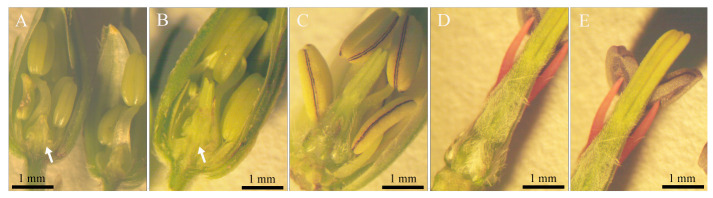
Pistil development. (**A**) The carpels of the young pistil had just met. (**B**) The pistil with stigmas was lower than the short stamens. (**C**) The pistil exceeding the short stamens, with trichomes at the base of the ovary. (**D**,**E**) A pistil exceeding the long stamens, with trichomes at the ovary and style (same flower in Figure 8D,E). White arrows indicate pistils.

**Figure 11 plants-12-03178-f011:**
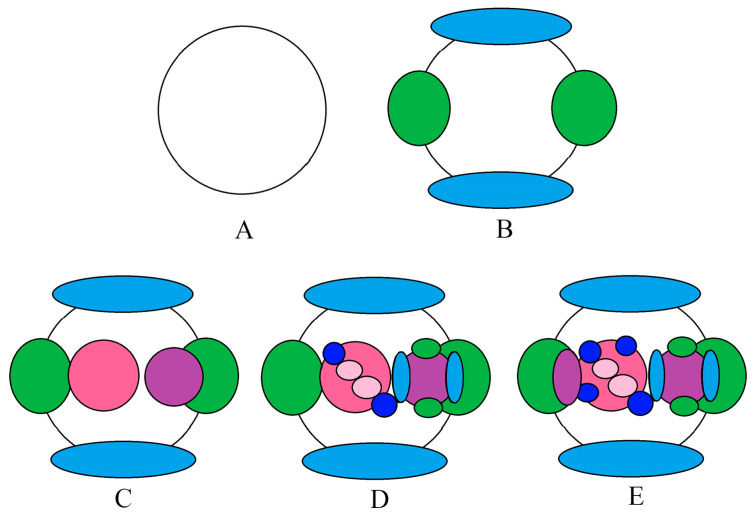
Developmental process of the dichasial inflorescence of *G. koreanum*. (**A**) The apical meristem initially appeared in a flat form. (**B**) The stipular primordia and the leaf primordia formed. (**C**) The floral bud primordium and the terminal bud primordium began to differentiate. (**D**) The floral bud primordium began to form two bract primordia and two flower buds, and the terminal bud primordium began to form the new stipule and leaf primordia. (**E**) The lateral bud primordium began to differentiate and the other two bract primordia on the floral bud formed. Blue represents the stipular primordia; green represents the leaf primordia; dark pink represents the small inflorescence primordia; purple represents the terminal primordia and the lateral bud primordia; light pink represents the two flower primordia; dark blue represents the bract primordia.

**Figure 12 plants-12-03178-f012:**
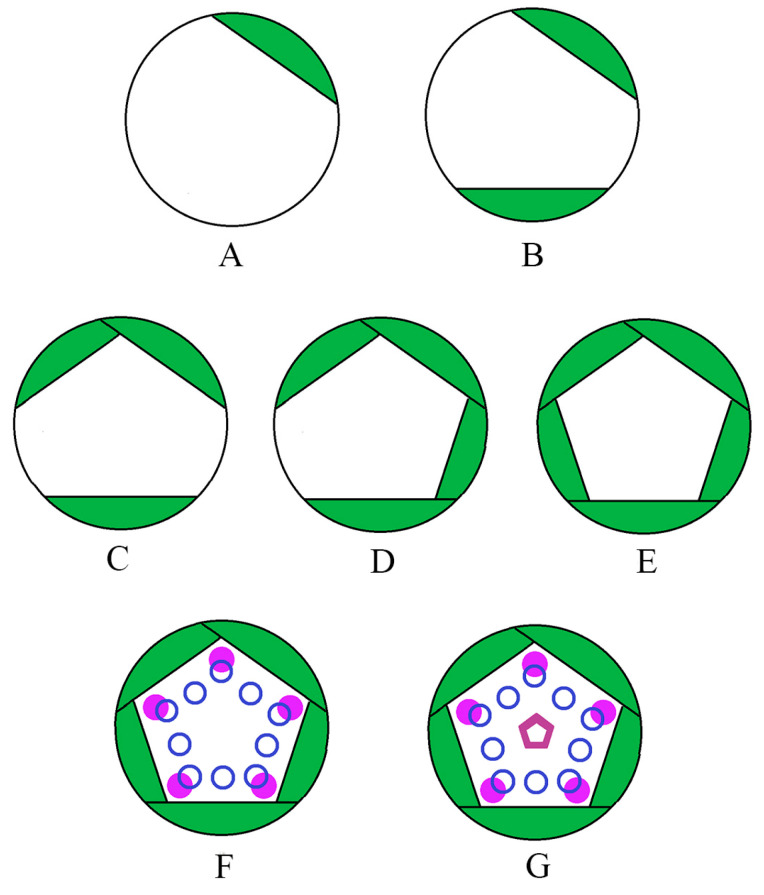
Developmental process of *G. koreanum* flower. (**A**) First sepal formed. (**B**) Second sepal formed. (**C**) Third sepal formed. (**D**) Fourth sepal formed. (**E**) Fifth sepal formed. (**F**) A total of 5 petal primordia and 10 stamen primordia formed. (**G**) Pistil primordia formed. Green represents the sepal primordia; the pink solid circles represent the petal primordia; the blue purple hollow circles represent the stamen primordia; the hollow pentagon represents the pistil primordium.

## Data Availability

Data sharing is not applicable to this article.

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
