# Peer review of "Development of Flowers Buds and Mixed Buds in the Dichasial Inflorescence of Geranium koreanum Kom. (Geraniaceae)"

_plants, 2023, doi:10.3390/plants12183178_

Round 1

Reviewer 1 Report

The subject of the work is very interesting and is addressed in a thorough manner.

The discussion analyzes the state of the art well and compares the available data with what will be studied about the species G. koreanum.

The methodology is clearly presented and is easy to compare with similar cases.

The presentation of the results clarifies every detail of bud development, both in flowers and mixed.

The discussion delves into the comparison with similar studies on other species, highlighting the advantages of the methodology used.

The overall language of the text is clear and smooth. I recommend a quick review, for safety. For example, in line 294 it should be "the plant needs" instead of "the plant need"

Overall, an excellent job

The overall language of the text is clear and smooth. I recommend a quick review

Author Response

Dear reviewer,

Thank you very much for taking the time to review the manuscript for us. Thank you very much for your feedback!

This revised version mainly includes the following modifications:

  1. The language has been revised throughout the text;
  2. Revised the titleï¼›
  3. Added content on the pattern of inflorescence (2.1)
  4. Figures 2, 3, 4, and 6 have added or replaced pictures, and corresponding text content has been modified to some extent.
  5. added arrows on the image.

Please review our revised manuscript again. Looking forward to your reply.

Best regards,

Reviewer 2 Report

My comments are below in the PDF file.

The language of the article requires moderate revision by a native speaker.

Author Response

Dear reviewer:

Thank you very much for taking the time to review the manuscript for us. Thank you very much for your feedback!

Based on your comment, we have made revisions to the article item by item. 

Please refer to the attachment for detailed modifications

Thank you again for your careful review. If there are any problem please let us know and we will continue to make the modifications.

Best regards,

Round 2

Reviewer 2 Report

Dear authors,

thak you very much for a carefull revision and editing of the manuscript  based on my recommendations. The adjustments made in relation to the pictures and schemes have improved the overall quality of the article. Let me express my appreciation about the scheme Figure 1C, is well elaborated, clear and easy to understand.

I am satisfied with your responses to my recommendations.

Note to the paragraf number 15 – scientific name: The full Latin name (scientific name) of the species must include the name of the authors e.g. the complete name of your observed species is Geranium koreanum Kom. (source – IPNI International Plant Names Index). Please edit the title of the article as follows: Development of flowers buds and mixed buds in the dichasial inflorescence of Geranium koreanum Kom. (Geraniaceae)

Note to the paragraph number 17: It appears that the development of the buds and inflorescence of Geranium is a complicated phenomenon, I advice you to continue the observation.

Note to the paragraph number 18: Thank you for your explanation. I accepted it.

Note to the paragraph number 20: Thank you for your editing. I accepted it.

Note to the paragraph number 30: As I mentioned above, scheme is very graphic and understandable.

Note to the paragraph number 31: The edited photo is illustrative enough. Thank you for editing.

In Figure 2 (B,E,F) there is grammatical mistake in term „stpule“ Please correct it to „stipule“.

I fully agree with rest of your feedback.

I hope my contributions was helpfull.

Best regards,

The quality of English language is appropriate. Only minor grammar mistakes need to be corrected.

Author Response

Dear reviewer,

Thank you very much for your feedback and suggestions, which are very meaningful and helpful.

In this revised version, we have revised the title and replaced Figure 2. We have also proofread the English language of the entire text, and corrected some minor grammar errors.

We will continue to study the inflorescence development of Geranium koreanum, which is an interesting project.

If there are any problem, please let us know and we will continue to make the modifications.

Best regards,